# Monkeypox and Its Possible Sexual Transmission: Where Are We Now with Its Evidence?

**DOI:** 10.3390/pathogens11080924

**Published:** 2022-08-17

**Authors:** Ranjit Sah, Abdelaziz Abdelaal, Abdullah Reda, Basant E. Katamesh, Emery Manirambona, Hanaa Abdelmonem, Rachana Mehta, Ali A. Rabaan, Saad Alhumaid, Wadha A. Alfouzan, Amer I. Alomar, Faryal Khamis, Fadwa S. Alofi, Maha H. Aljohani, Amal H. Alfaraj, Mubarak Alfaresi, Jumana M. Al-Jishi, Jameela Alsalman, Ahlam Alynbiawi, Mohammed S. Almogbel, Alfonso J. Rodriguez-Morales

**Affiliations:** 1Tribhuvan University Teaching Hospital, Institute of Medicine, Kathmandu 44600, Nepal; 2Harvard Medical School, Boston, MA 02115, USA; 3School of Medicine, Boston University, Boston, MA 02118, USA; 4Tanta Research Team, Tanta 31527, Egypt; 5MBBCh, Faculty of Medicine, Tanta University, Tanta 31527, Egypt; 6Faculty of Medicine, Al-Azhar University, Cairo 11651, Egypt; 7College of Medicine and Health Sciences, University of Rwanda, Kigali 3286, Rwanda; 8MBBCh, Faculty of Medicine Fayoum University, Fayoum 63514, Egypt; 9National Public Health Laboratory, Kathmandu 44600, Nepal; 10Molecular Diagnostic Laboratory, Johns Hopkins Aramco Healthcare, Dhahran 31311, Saudi Arabia; 11College of Medicine, Alfaisal University, Riyadh 11533, Saudi Arabia; 12Department of Public Health and Nutrition, The University of Haripur, Haripur 22610, Pakistan; 13Administration of Pharmaceutical Care, Al-Ahsa Health Cluster, Ministry of Health, Al-Ahsa 31982, Saudi Arabia; 14Department of Microbiology, Faculty of Medicine, Kuwait University, Safat 13110, Kuwait; 15Microbiology Unit, Department of Laboratories, Farwania Hospital, Farwania 85000, Kuwait; 16Department of Clinical Laboratory Sciences, College of Applied Medical Sciences, Imam Abdulrahman Bin Faisal University, Dammam 31441, Saudi Arabia; 17Infection Diseases Unit, Department of Internal Medicine, Royal Hospital, Muscat 1331, Oman; 18Department of Infectious Diseases, King Fahad Hospital, Madinah 42351, Saudi Arabia; 19Pediatric Department, Abqaiq General Hospital, First Eastern Health Cluster, Abqaiq 33261, Saudi Arabia; 20Department of Pathology and Laboratory Medicine, Sheikh Khalifa General Hospital, Umm Al Quwain P.O. Box 499, United Arab Emirates; 21Department of Pathology, College of Medicine, Mohammed Bin Rashid University of Medicine and Health Sciences, Dubai 505055, United Arab Emirates; 22Internal Medicine Department, Qatif Central Hospital, Qatif 35342, Saudi Arabia; 23Infection Disease Unit, Department of Internal Medicine, Salmaniya Medical Complex, Ministry of Health, Kingdom of Bahrain, Manama 435, Bahrain; 24Infectious Diseases Section, Medical Specialties Department, King Fahad Medical City, Riyadh 12231, Saudi Arabia; 25Department of Medical Laboratory Sciences, College of Applied Medical Sciences, University of Hail, Hail 4030, Saudi Arabia; 26Grupo de Investigación Biomedicina, Faculty of Medicine, Fundacion Universitaria Autonoma de las Americas, Pereira 660003, Risaralda, Colombia; 27Institucion Universitaria Vision de las Americas, Pereira 12998, Risaralda, Colombia; 28Master of Clinical Epidemiology and Biostatistics, Universidad Cientifica del Sur, Lima 15024, Peru

**Keywords:** sexual transmission, monkeypox, emerging, global, epidemic

## Abstract

Monkeypox is a rare disease but is increasing in incidence in different countries since the first case was diagnosed in the UK by the United Kingdom (UK) Health Security Agency on 6 May 2022. As of 9 August, almost 32,000 cases have been identified in 89 countries. In endemic areas, the monkeypox virus (MPXV) is commonly transmitted through zoonosis, while in non-endemic regions, it is spread through human-to-human transmission. Symptoms can include flu-like symptoms, rash, or sores on the hands, feet, genitalia, or anus. In addition, people who did not take the smallpox vaccine were more likely to be infected than others. The exact pathogenesis and mechanisms are still unclear; however, most identified cases are reported in men who have sex with other men (MSM). According to the CDC, transmission can happen with any sexual or non-sexual contact with the infected person. However, a recent pooled meta-analysis reported that sexual contact is involved in more than 91% of cases. Moreover, it is the first time that semen analysis for many patients has shown positive monkeypox virus DNA. Therefore, in this review, we will describe transmission methods for MPXV while focusing mainly on potential sexual transmission and associated sexually transmitted infections. We will also highlight the preventive measures that can limit the spread of the diseases in this regard.

## 1. The Epidemiology of Monkeypox

In 1958, Danish virologist Preben Christian Alexander von Magnus discovered the monkeypox virus (MPXV) in Copenhagen [1]. Monkeypox (MPX) was first reported as a zoonotic disease that is transmitted from animals to humans by MPXV. It is a double-stranded DNA virus belonging to the Orthopox genus of the Poxviridae family. The Orthopox genus also includes cowpox, smallpox, and rabbitpox. MPX was first reported in monkeys, although it is believed that Rodents are the natural host [2]. Given the similarity between both monkeypox and smallpox, smallpox vaccines offer cross-protection against MPXV. There are two types of MPXV: the Central African (Democratic Republic of Congo) clade and the West African clade (now mainly called clades 1 and 2). The case-fatality rate of the Central African clade is significantly higher than that of the West African type [2,3]. MPX is transmitted zoonotically by direct contact with body fluids, blood, or lesions of infected animals; therefore, it was reported as an occupational disease [2]. It was listed by the World Health Organization (WHO) as having epidemic and pandemic potential [4].

The first case of human infection was reported on 1 September 1970. A nine-month-old child was admitted to a Democratic Republic of Congo hospital. He presented with a fever and rash that had developed two days prior to presentation. Upon examination, hemorrhagic lesions with centrifugal distribution were noted [5]. The patient was suspected to have smallpox, but genome sequences confirmed the presence of both the West and Central African clades of MPXV [6]. MPX has since spread to several African countries and remained endemic in Central and West Africa, including countries such as the Democratic Republic of Congo (DRC), Central African Republic, Nigeria, and Liberia, among others [6].

Human-to-human transmission of MPX emerged after the 1996–1997 Democratic Republic of Congo outbreak [7]. This type of transmission was mainly mediated through direct contact with rash, scabs, sores, and scabs of infected individuals. Moreover, disease transmission can occur through droplet infection and through contact with objects and surfaces contaminated by infected cases [2]. However, in the recent multi-country MPX outbreak in 2022, the Centers for Disease Control and Prevention CDC declared that transmission might also occur by hugging, kissing, or contact with the anus or genitals of individuals in addition to oral, anal, and vaginal sex [8]. It was also linked to the discontinuation of the smallpox vaccine, weakened immunity, and poor living standards [7]. The smallpox vaccine was shown to be 85% effective in preventing MPX. It has an incubation period ranging from 5–21 days with a similar clinical presentation to that of smallpox [6]. In the same context, smallpox vaccines have been shown to be 85% effective in preventing MPX [9].

Monkeypox, a rare disease in the past, has increased in incidence in different countries since the reported case in the UK was diagnosed by the United Kingdom (UK) Health Security Agency on 6 May 2022. Since then, there have been several cases reported in Spain, Portugal, the UK, and the Americas. Although there is no evident epidemiological link between reported cases, current preliminary evidence highlights that the current multi-country outbreak is closely related to the previous Central African clade based on the genetic analysis of the current MPXV [10]. In addition, the geographical dispersal is much greater than that of past outbreaks. Therefore, and because of the increased burden and number of cases, on 23 July 2022, the WHO declared MPX a public health emergency of international concern. During the first week of the epidemic, suspected and confirmed cases were reported in 24 countries, some of which had travel links to Europe [11]. As of 9 August 2022, the CDC had reported 31,800 confirmed MPX cases in 89 countries, with the majority of cases being reported in countries that have not historically reported MPX (82 out of 89 countries) [12]. Almost one-third of reported cases (9492 cases) are located in the US followed by Spain, Germany, and the UK, respectively.

## 2. Diagnosis and Clinical Presentation of Monkeypox

Suspected cases are defined as (1) cases with sudden symptom onset of rash with or without anogenital complaint that is unexplained by other diseases and (2) the presence of one or more of the following symptoms: fever of sudden onset, myalgia, headache, backache, asthenia, or lymphadenopathy [13]. A probable case is reported as one that meets the suspected criteria of having a positive Orthopoxvirus DNA test by PCR. A confirmed case is laboratory-confirmed after the confirmation of MPXV presence in collected samples through PCR. Cases are excluded from the above classification if they did not develop a rash within five days, their specimens showed negative test results, their tests showed an absence of antibodies, or if a differential diagnosis is confirmed [14]. 

The US CDC reported that MPX infection has an incubation period of 1–2 weeks before the appearance of initial symptoms, with infection usually lasting 2–4 weeks. The severity of infection depends on the patient’s prior health, mode of infection, and the strain of the virus. In the past, the Central African clade has shown more severe signs and symptoms with a higher mortality rate when compared to the West African clade [15]. However, in the recent multi-country outbreak, more research is still needed to highlight the exact presentation patterns, associated characteristics, and exact mortality rate.

Based on recent data, the clinical manifestations of MPX are categorized based on timing into (1) invasion period and (2) skin manifestations. The invasion period includes flu-like symptoms such as headaches, fever, malaise, chills, and lymphadenopathy during the prodromal period. Skin manifestations are non-pleomorphic and are observed 1–3 days later, appearing in the sequential stages of macules, papules, vesicles, pustules, crust, and scars. Skin lesions are typically painful and last 2–4 weeks before completely healing and falling off [2]. MPX rash mainly presents as vesicles or pustules that are firm, deep-seated, well-circumscribed, and often develop umbilically. These lesions are contagious from their onset till the formation of a scab and are commonly confused for chickenpox or smallpox [14]. Skin pigmentation and scars may remain after healing. 

The most common signs and symptoms include fever (reported in 54.29% of cases), inguinal lymphadenopathy (45.71%), exanthema (40%), ulcers and vesicles on the genitals and anal region (31.43%), asthenia and fatigue (22.86%), headache (25.71%), myalgia (17.14%), cervical lymphadenopathy (11.43%), and axillary lymphadenopathy (5.71%) [1]. Some patients reported having a rash followed by other symptoms. Meanwhile, others have reported the occurrence of a rash without prodromal symptoms; however, the rash was followed by systemic symptoms/signs [8]. MPX-infected individuals present with fever before the rash, while lymphadenopathy presents at the time of fever or rash occurrence or 1–2 days after the beginning of rash. Additionally, it has been highlighted that MPX cases in the current multi-country outbreak tend to exhibit exclusive genital lesions, which extinguish the presentation of the current MPX from previous outbreaks [16].

Typically, the rash first appears on the face and then spreads to other areas of the body; however, many studies reported an atypical presentation, with the most common initial manifestation being on the perianal or genital region, followed by a monomorphic and centrifugal distribution [1,14]. Based on data from reported cases in the US, it was noted that the most common sites for rash occurrence were the arms followed by the trunk, leg, face, hand, perianal, oral, neck, genital, and feet, respectively [14]. It has been reported that the presence of herald lesions at the point of sexual contact and the absence of prodromal symptoms are suggestive of sexual disease transmission [17]. One Australian case reported a superinfection presenting bacterial cellulitis of the penile shaft and lower central abdomen [1]. 

## 3. Transmission Methods of Monkeypox Virus

### 3.1. Non-Sexual Transmission

MPX disease could be transmitted either from zoonosis or through human-to-human transmission. From 1970 to 2003, the endemic areas for MPX were the rainforest of Africa [18]. However, in 2003, the outbreak of MPX was first identified outside the endemic regions in Africa and appeared in the US. Since then, sporadic outbreaks have spread worldwide, all linked to travel to endemic areas in Africa [18]. Since MPXV is an orthopoxvirus, it could be transmitted through zoonosis. The same animal reservoir is still unclear, since monkeys (the natural reservoir for the virus) were eliminated [18]. A wide variety of species have been reported to be susceptible to MPXV. These include tree and rope squirrels, Gambian marsupials, monkeys, and non-human primates (NHPs), as well as other species [12,17,18]. Additionally, MPX disease is also common among rodents in Africa [13]. Transmission could be through direct contact with body fluid or blood or MPX lesions of infected animals. Moreover, inadequately cooked meat could transport the disease [2]. 

According to the CDC, an individual can become infected with MPXV in different ways related to sexual or non-sexual contact. For non-sexual contact, it could be transmitted through: (1) direct contact with the rash, sores, or scabs of infected individuals, (2) contact with objects and surfaces that are contaminated by infected individuals, and (3) through respiratory droplets or oral fluids from an infected individual. In the recent multi-country outbreak of MPX, the first documented case in the UK reported a history of traveling to endemic areas (Nigeria). Moreover, in Portugal, three patients had contact with animals (two patients with cats and one with pigs), while another reported contact with another patient [1]. However, there is no epidemiological link between a large number of cases of MPXV and Central or West Africa [18,19]. That may indicate that MPXV spread among people for a prolonged period through human-to-human transmission without detection [19]. That being said, human-to-human transmission of MPX has been documented to occur among household contacts and shared housing inhabitants (e.g., in prisons) as well as among health care providers who have direct, close, and sustained contact with MPX patients [14].

### 3.2. Potential Sexual Transmission

Although evidence suggests that at-risk populations of MPXV transmission are people with a history of close physical contact with a symptomatic person, evidence has shown another mode of possible transmission. Available data indicate that MPXV cumulative cases have mostly been documented among men who have sex with men (MSM) [17]. The WHO has described sexual contact or multiple or different sexual partners in the 21 days before symptom onset as one risk factor. That asks the question regarding the possible sexual transmission of this disease. For instance, in a study reporting 54 MPX cases presenting at one health center in the UK, it was noted that all cases (100%) were MSM [20]. This is similar to the findings of a study conducted in Italy, where four documented MSM MPX cases reported having sexual intercourse without using a condom [21]. Sexual contact could be the most important risk factor for MPXV in specific populations, particularly among MSM, resulting in a staggering number in line with this category. A recent review showed that “sexual exposure” could be reported in 91.67% of cases (n = 124) [1]. That was particularly evident in unprotected sexual intercourse with multiple, random, or anonymous partners [1]. These findings, although based on preliminary data, highlight that sexual intercourse could be an important route of disease transmission. This was confirmed in a recently published multi-center study by Thornhill et al. [22] that included 528 infected MPX cases from 16 countries during the period 27 April–24 June 2022. The study highlighted that 98% of individuals with confirmed infections were gay or bisexual. The authors suspected that the majority of cases became infected through sexual activity (95% of infected cases). Of note, 32 individuals underwent semen analysis and among them, 29 cases showed positivity for MPXV. That being said, sexual transmission of MPXV should be confirmed in larger studies that focus on assessing viral infectivity and replication in cell cultures, using isolated viruses from seminal samples.

In addition to existing symptoms of MPXV, the recently documented manifestations of the disease indicate signs of sexual transmission. Some lesions were found on the genitalia of infected individuals, such as deep-seated and well-circumscribed lesions [21]. Other perianal lesions included itching papules and inguinal lymphadenopathy. Interestingly, samples obtained from genital and anal lesions revealed positive PCR results as samples collected from the skin, feces, seminal fluid, nasopharynx, serum, and plasma in MPXV DNA in real-time PCR. MSM was the most commonly identified population to be significantly more affected by MPXV. That is supported by the recent evidence highlighting that more than 80% of cases were either gay or bisexual [23]. 

Another vital point to consider is the concomitant infection of MPXV with other sexually transmitted infections (STIs). This was observed among MSM cases that had MPX infection. For instance, all MSM cases diagnosed with MPXV had previous STIs, hepatitis C, syphilis, or human immune-deficiency virus (HIV) [21]. The association between MPX and HIV has been documented in endemic zones. For example, a study conducted in Nigeria found a correlation between HIV patients with MPXV [24]. This study revealed that coinfection of HIV type 1 and MPXV had larger lesions in the skin and genital ulcers than negative HIV cases. Interestingly, the recent CDC (USA) results reported a possible coinfection of MPXV and STIs. Furthermore, during the current outbreak, there were patients with MPX who also had chlamydia, herpes syphilis, and gonorrhea infections [25]. Importantly, in the recent study by Thornhill et al. [22], it was noted that 41% of MPX-infected cases (216 out of 528 cases) had concomitant HIV during the time of presentation, of which 95% were on antiretroviral therapy.

As mentioned above, MPXV could be transmitted through sexual activity. That is supported by laboratory analysis that found positive MPXV qPCR in the seminal fluid of MSM in Germany and other countries [21]. Similarly, a study conducted in Italy has supported a possible association between MPXV and sexual transmission. The INMI Monkeypox group showed that clinical features, including anal and genital lesions, were the first to appear among MSM, whose semen analysis showed MPXV DNA [21]. 

Available data, based on the clinical presentation of MPX, the site of occurrence of MPX lesions, the history of sexual activity among MSM, and the concurrent infection with STIs, support the transmission of MPX sexually. Further investigations are still warranted to confirm this observation. That being said, it should be noted that this disease can be transmitted through other methods which should not be neglected since successful prevention strategies should target all methods of disease transmission. 

## 4. Transmission-Related Preventive Measures

Health experts usually recommend various approaches to contain the dissemination of an outbreak, especially ones that affect thousands of individuals and communities worldwide and cause serious health issues to infected patients. In this context, various approaches have been proposed to limit MPXV transmission and reduce the number of infected patients. These include implementing quarantine measures (aiming at decreasing the risk of animal-to-human and human-human-to-human transmission), vaccination, and educational programs that aim to enhance individual-based awareness about the best practices to intervene against MPXV infection. We will expand on these approaches in the following sections.

### 4.1. Reducing Transmission

During outbreaks, healthcare authorities must seek the identification of and provide surveillance of new cases. In addition, evidence indicates that a significant risk factor to consider during MPV outbreaks is close contact with infected MPX cases. Therefore, healthcare experts have provided different prevention approaches that the public should commit to, including (1) the quarantine of infected patients from healthy individuals, (2) using disinfectants that are Environmental Protection Agency–registered, and (3) using appropriate personal protective equipment and practicing good personal hygiene to provide the best intervention, especially for household members that are in direct contact with infected cases [14].

In addition, household members and healthcare workers (HCWs) who are in direct or close contact with infected cases are at high risk of being infected. Therefore, frontline HCWs (usually in direct contact with suspected and confirmed MPX patients or others checking them for specimens) and household members must commit to standard infection control measurements. Furthermore, it is suggested that individuals previously vaccinated against smallpox should be selected to deal with the current cases. Moreover, quarantine implementation and the screening of individuals traveling from endemic communities are essential. Accordingly, some evidence suggests that such individuals and suspected cases should be quarantined for three weeks [2]. However, no clear guidelines were provided in the literature in this context.

Evidence shows that animal-to-human transmission has been responsible for many cases of MPV infections. Therefore, direct contact with wild confirmed/suspected cases of infected animals should be prohibited. Avoiding contact with blood, meat, or other body parts, whether the animal is dead or alive, is encouraged. Furthermore, animal meat should be adequately cooked before eating. 

Moreover, the quarantine of infected animals is encouraged to prevent transmitting the infection to other animals and humans. A 30-day quarantine should also be provided for animals having close contact with other confirmed animal/human cases. These animals should be handled according to standardized infection control measures and symptoms should be observed during the quarantine period [17]. In addition, further studies are encouraged to outline the reasons behind the outbreaks in non-endemic areas. These studies should be directed to investigate the potential sources of infection and reduce the intensity and spread of the disease.

### 4.2. Vaccination

No vaccines have been developed for MPV infection. However, past evidence indicates that smallpox vaccines can prevent the spreading of MPXV infections due to cross-reaction, with an estimated efficacy of 85% [15]. Accordingly, two smallpox vaccines have been approved for preventing MPV, including the second-generation smallpox vaccine ACAM2000, approved by the Food and Drug Administration for post-exposure prophylaxis, and the third-generation JYNNEOS^TM^, approved for different population groups.

Various investigations indicated the efficacy of these vaccines [26,27,28,29,30,31,32,33]. However, the central dilemma in using them lies with their associated adverse events. For instance, some studies reported that ACAM2000 should not be administered to patients with atopic dermatitis and those with vulnerable immune systems (including patients with recent stem cell transplantations and others with immunocompromising conditions or receiving immunosuppressants) [33,34,35,36]. On the other hand, JYNNEOS^TM^ has a good safety profile with proven efficacy in animal and human studies. It is a non-replicating vaccine with low immunological reactogenicity to different generations of smallpox vaccines [36,37,38,39,40,41].

Moreover, the response of public health authorities is essential to help contain such outbreaks. For instance, the CDC has provided the availability of post-exposure prophylaxis vaccines for contacts with intermediate (after contacting an unmasked patient for ≥3 h within ≤6 ft without wearing a mask) and high-risk exposures. In addition, the CDC recommends that vaccination be administered within four days of the time of exposure [42]. On the other hand, post-exposure prophylaxis should not be dispensed to individuals with uncertain or low-risk profiles (such as HCWs contacting a patient without eye protection) [43].

Furthermore, ring vaccination of MPX contacts offers a great benefit in reducing transmission of MPXV given its ability to cut off the chain of virus transmission in addition to preventing the occurrence of severe MPX [44]. Efforts are now being made to effectively initiate trials on ring vaccination among MPX contacts [45]. Applying ring vaccination is also encouraged. The approach was successfully used in previous outbreaks, including smallpox and Ebola virus [46,47,48]. It is achieved by vaccinating family members/households and members with the parameters of a confirmed case. In addition, the strategy should include contacts and contacts of contacts to infected patients, irrespective of their geographical distribution. This might not be easy to apply and providing vaccines might not be feasible for these individuals. Therefore, contact tracing through a systematic approach, early identification and the quarantine of cases, and strict application of other prevention approaches are essential, especially when vaccines cannot be offered. These measurements are encouraged due to the stoppage of smallpox vaccination [49,50]. This poses an additional challenge for healthcare authorities to deliver vaccines on a wide scale beside the challenge of increasing awareness and vaccine acceptance among the general population [51,52,53].

That being said, there is an urgent need to properly disseminate vaccination strategies worldwide in a timely manner before the burden of MPX becomes uncontainable. Unfortunately, vaccine supplies are currently limited, given the fact that they are provided by the military only, and this constitutes a major challenge in the control of MPX spread. Therefore, all efforts should be united to effectively execute a proper vaccination plan worldwide as soon as possible to minimize the infection rate.

### 4.3. Others

Raising awareness for the general public and individuals having close contact with MPXV cases is also essential to enhance prevention and reduce transmission of MPXV. Accordingly, different educational campaigns can be provided. For instance, training healthcare individuals responsible for dealing with confirmed/suspected cases and others accountable for collecting specimens is mandatory. Furthermore, safe storage, handling, and transportation of human/animal specimens should be performed with triple packaging according to the WHO guidance for transporting infectious substances [54]. Moreover, enhancing the knowledge (about the symptoms, signs, transmission, and prevention) and attitude (towards sticking to the different prevention approaches and seeking medical care whenever needed) among the public and highly vulnerable groups should be provided.

It is essential to consider the stigma caused by the increasing pressure, especially on the LGBTQI+ community, secondary to the growing claims that many confirmed MPXV cases are individuals who identify themselves as “men who have sex with men” [7]. However, as we mentioned in an earlier section, sexual transmission is not the only route of transmitting the infection. Furthermore, close physical contact (including sexual transmission) is not limited to these communities, irrespective of gender identity and sexual orientation. Therefore, although sexual transmission is a significant route for transmitting the infection, it should not be specified as the only route, similar to what happened in the 1980s during the HIV outbreaks [12,54], which was associated with severe healthcare concerns and worsened the situation. Therefore, public and healthcare authorities should provide measures to overcome this stigma.

Indeed, MSM individuals would want to be properly treated, if infected, and/or vaccinated if they had been in contact with a confirmed case. Therefore, awareness campaigns should be implemented and MSM individuals should be encouraged to seek health care if they exhibit symptoms or become exposed to the disease [44].

Accordingly, public communications should indicate that the disease is not limited to specific communities and is a global public health concern that requires the integration of the public and commitment to the different prevention approaches, rather than generating a gender-based stigma, which can only make matters worse [54]. On the other hand, it is also practical not to neglect educational campaigns for these groups, which should also increase their awareness and attitude towards the infection. For instance, educational campaigns should be directed to seeking and following healthcare experts’ prevention and treatment recommendations. Moreover, avoiding both sexual intercourse and being in close contact with confirmed/suspected cases should be prioritized [17].

## 5. Conclusions

MPXV is now widespread in Europe and different countries across the world. It has different modes of transmission, with sexual contact considered an important mode of transmission. Men who have sex with other men are more liable to become infected than others. Recent evidence also highlights an association between MPX and STIs such as HIV, which warrants further investigations to carefully assess the disease’s burden.

Furthermore, efforts should be directed towards limiting the spread of the disease, not just by case tracking, quarantine implementation, and vaccination, but also through controlling the contributing factors to disease transmission, either sexually or non-sexually. This can be tackled by providing educational programs for the public regarding transmission methods, limiting the stigma associated with identifying sexually transmitted cases, and providing proper management for such cases. Finally, we are on the verge of an MPX pandemic. Therefore, all efforts and available resources should be directed towards investigating this disease and its preventive measures to contain it before it affects millions, as occurred with COVID-19.

## Data Availability

Not applicable.

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
