# Peer review of "Monkeypox and Its Possible Sexual Transmission: Where Are We Now with Its Evidence?"

_pathogens, 2022, doi:10.3390/pathogens11080924_

Round 1

Reviewer 1 Report

The revised manuscript is good to go for me.

Best

Author Response

Reviewer 1 Comments:

Comment 1: The revised manuscript is good to go for me

Response: Thank you agreeing for publication of our article. We really appreciate for your valuable and impactful comments that has improved our manuscript.

Reviewer 2 Report

The manuscript by Ranjit et al summarized the reported information on monkeypox from different aspects. Since the UK case was reported on May 6, 2022, more than 28,000 cases have been reported as of 05 Aug 2022. The number of confirmed infected cases is increasing at a rapid rate, so WHO Director-General has declared the ongoing monkeypox outbreak a public health emergency of international concern. As described in this manuscript, the monkeypox mainly transmitted through non-sexual route in the past, such as direct contact with body fluid of infected cases, inadequately cooked meat of infected animals. During this outbreak, Sexual transmission seems to be the main route, especially by MSM. The authors thought that several methods should be taken to block the monkeypox transmission, such as quarantine, disinfection, ring vaccine which were the regular methods to prevent infectious diseases. Overall, this paper could give us some useful information to make plans on prophylaxis and treatment of monkeypox. However, there are some questions remaining to be addressed, especially the language and logic. Please see the following comments.

1.       This manuscript is not good-writing and lack of logic. Many descriptions in the main body are unreadable and will let readers feel confused, for example page 4, line 118-127, line 134-140, line 150-156, line 163-165; page 5, line 173-176, line 208-210; page 6, line 242-244, line 269-270; page 7, line 283-285, line 310-313, line 321-324; page 8, line 328-331. So, I think it needs extensive editing of English language and style.

2.       Page 1, line 51-52, I think Monkeypox was a rare disease in the past, but not now, and the UK case is not the first reported case in history. The description should be “Monkeypox which was a rare disease in the past is rising nowadays in different countries since the reported case in the UK was diagnosed on May 6, 2022, by the United Kingdom (UK) Health Security Agency”.

3.       Page 1, line 53, please check the data and update based on the USA CDC report (https://www.cdc.gov/poxvirus/monkeypox/response/2022/world-map.html).

4.       Page 2, line 76-78, based on the published papers, the central African type has a significantly higher case fatality ratio than the West African type. https://doi.org/10.1016/j.amsu.2022.103975, https://doi.org/10.1016/S1473-3099(22)00228-6.

5.       Page 2, line 97, where is the data 85% from? I think from this paper https://doi.org/10.1093/ije/17.3.643. Please cite.

6.       Page 2, line 100-102, I think there should be a citation to point out where the conclusion is from.

7.       Page 3 line 105-116, please check the data and update based on the above CDC report which is more authoritative than the referenced one. Line 116 should be “have reported at least one confirmed case of MPX”.

8.       Page 5, line 211-214, this study has been done, please see this paper and update this manuscript. https://doi.org/10.1016/S1473-3099(22)00513-8

9.       Page 7, line 294-297, This paragraph should not appear in this section, may move it to the Vaccination section.

10.   Page 7, line 315-317, this description doesn’t match the recommendation of USA CDC. CDC recommends that the vaccine be given within 4 days from the date of exposure for the best chance to prevent onset of the disease. https://www.cdc.gov/poxvirus/monkeypox/considerations-for-monkeypox-vaccination.html

Author Response

Reviewer 2 Comments:

The manuscript by Ranjit et al. summarized the reported information on monkeypox from different aspects. Since the UK case was reported on May 6, 2022, more than 28,000 cases have been reported ass of 05 Aug 2022. The number of confirmed infected cases is increasing at a rapid rate, so WHO Director-General has declared the ongoing monkeypox outbreak a public health emergency of international concern. As described in this manuscript, the monkeypox mainly transmitted through non-sexual route in the past, such as direct contact with body fluid of infected cases, inadequately cooked meat of infected animals. During this outbreak, sexual transmission seems to be the main route, especially MSM. The authors thought that several methods should be taken to block the monkeypox transmission, such as quarantine, disinfection, ring vaccination which were the regular methods to prevent infectious diseases. Overall, this paper could give us some useful information to make plans on prophylaxis and treatment of monkeypox. However, there are some questions remaining to be addressed, especially the language and logic. Please see the following comments.

Comment 1: This manuscript is not good-writing and lack of logic. Many descriptions in the main body are unreadable and will let readers feel confused, for example page 4, line 118-127, line 134-140, line 150-156, line 163-165; page 5, line 173-176, line 208-210; page 6, line 242-244, line 269-270; page 7, line 283-285, line 310-313; page 8, line 328-331. So, I think it needs extensive editing of English language and style.

Response: Thank you for highlighting these issues. One of the co-authors (AA) who is fluent in English revised the English structure of the manuscript and corrected all of the typos and grammatical mistakes that were found. All edits can be tracked through the “Track changes” option in word and they are also highlighted in yellow in the revised manuscript.

Point by point description of the changes made to the following paragraphs:

Page 4; lines 118-127: changed to:

Page 4; lines 134-140: changed to: “Suspected cases are defined as (1) cases with sudden symptom onset of rash with or without anogenital complaint that is unexplained by other diseases and (2) the presence of one or more of the following symptoms: fever of sudden onset, myalgia, headache, backache, asthenia, or lymphadenopathy [13]. A probable case is reported as one that meets the suspected criteria and has demonstrated the presence of MPXV by polymerase chain reaction (PCR), immunohistochemical test, electron microscopy, or has detectable levels of IgM antibodies within 4-56 days after rash presentation. A confirmed case is laboratory-confirmed after the confirmation of MPXV presence in collected samples through PCR. Cases are excluded from the above classification if they did not develop a rash within five days, their specimens showed negative test results, their tests showed an absence of antibodies, or if a differential diagnosis is confirmed [14].”

Page 4; lines 150-156: changed to: “Some patients reported having a rash followed by other symptoms. Meanwhile, others reported the occurrence of a rash without prodromal symptoms; however, the rash was followed by systemic symptoms/signs [8]. MPX-infected individuals present with fever before the rash, and lymphadenopathy presents at the time of fever or rash occurrence or 1-2 days after the beginning of rash. Additionally, it has been highlighted that MPX cases in the current multi-country outbreak tend to exhibit exclusive genital lesions, which extinguishes the presentation of the current MPX from previous outbreaks [16].”

Page 4; lines 163-165: changed to: “It has been reported that the presence of herald lesions at the point of sexual contact and the absence of prodromal symptoms are suggestive of sexual disease transmission [17].”

Page 5; lines 173-176: changed to: “Since MPXV belongs to orthopoxviruses, it could be transmitted through zoonosis. The same animal reservoir is still unclear; since monkeys (the natural reservoir for the virus) were eliminated [18].”

Page 5; lines 208-210: changed to: “That was evident, particularly, in unprotected sexual intercourse with multiple, random, or anonymous partners [1]. These findings, although based on preliminary data, highlight that sexual intercourse could be an important route of disease transmission.”

Page 6; lines 242-244: changed to: “Available data, based on the clinical presentation of MPX, the site of occurrence of MPX lesions, the history of sexual activity among MSM, and the concurrent infection with STIs, support the transmission of MPX sexually. Further investigations are still warranted to confirm this observation. That being said, it should be noted that this disease can be transmitted through other methods which should not be neglected since successful prevention strategies should target all methods of disease transmission.”

Page 6; lines 269-270: changed to: “Besides, household members and healthcare workers (HCWs) who are in direct or close contact with infected cases are at high risk of being infected.”

Page 7; lines 283-285: changed to: we deleted this paragraph for the lack of applicability during the current PHEIC.

Page 7; lines 310-313: changed to: “On the other hand, JYNNEOSTM has a good safety profile with proven efficacy in animal and human studies. It is a non-replicating vaccine with low immunological reactogenicity to different generations of smallpox vaccines [36-41].”

Page 8; lines 328-331: changed to: “These measurements are encouraged due to the stoppage of smallpox vaccination [47, 48]. This poses an additional challenge for healthcare authorities to deliver vaccines on a wide scale besides the challenge of increasing awareness and vaccine acceptance among the general population [49-51].”

Comment 2: Page 1, line 51-52, I think monkeypox was a rare disease in the past, but not now, and the UK case is not the first reported case in history. The description should be “Monkeypox which was a rare disease in the past is rising nowadays in different countries since the reported case in the UK was diagnosed on May 6, 2022 by the United Kingdom (UK) Health Security Agency”.

Response: Thank you for your comment. We added this sentence at page 4, lines 114-116 as follows: “Monkeypox which was a rare disease in the past is rising nowadays in different countries since the reported case in the UK was diagnosed on May 6, 2022 by the United Kingdom (UK) Health Security Agency. Since then,….”

Comment 3: Page 1, line 53, please check the data and update based on the USA CDC reported (https://www.cdc.gov/poxvirus/monkeypox/response/2022/world-map.html).

Response: Thank you for the great suggestion. We edited this part as suggested and updated the number of confirmed cases as follows: “As of August 9, 2022, the CDC has reported 31800 confirmed MPX cases in 89 countries, with the majority of cases being reported in countries that have not historically reported MPX (82 out of 89 countries) [9]. Almost one third of reported cases are situated in the US followed by Spain, Germany, and the UK, respectively.”

Comment 4: Page 2, line 76-78, based on the published papers, the central African type has significantly higher case fatality ratio than the West African type. https://doi.org/10.1016/j.amsu.2022.103975, https://doi.org/10.1016/S1473-3099(22)00228-6.

Response: Thank you for your recommendation. We edited this part as requested as follows: “The case-fatality rate of the central African clade is significantly higher than the West African type [2, 3].”

Comment 5: Page 2, line 97, where is the data 85% from? I think from this paper https://doi.org/10.1093/ije/17.3.643. Please cite.

Response: Thank you for your suggestion. We added the recommended citation.

Comment 6: Page 2, line 100-102, I think there should be a citation to point out where the conclusion is from.

Response: Thank you for your comment. This statement was driven from the WHO’s update statement on monkeypox [https://www.who.int/emergencies/disease-outbreak-news/item/2022-DON396].

Comment 7: Page 3, line 105-116, please check the data and update based on the above CDC report which is more authoritative than the referenced one. Line 116 should be “have reported at least one confirmed case of MPX”.

Response: Thank you for your suggestion. This edit was applied as per the suggestion in Comment #3 as follows: “As of August 9, 2022, the CDC has reported 31800 confirmed MPX cases in 89 countries, with the majority of cases being reported in countries that have not historically reported MPX (82 out of 89 countries) [11]. Almost one third of reported cases are situated in the US followed by Spain, Germany, and the UK, respectively.”

Comment 8: Page 5, lines 211-214, this study has been done, please see this paper and update this manuscript, https://doi.org/10.1016/S1473-3099(22)00513-8.

Response: Thank you for your comment. This part has been updated as recommended.

Comment 9: Page 7, line 294-297, this paragraph should not appear in this section, may move it to the Vaccination section.

Response: Thank you for your comment. We moved the “ring vaccination” part to the “vaccination” section [page 7, lines 317-320].

Comment 10: Page 7, line 315-317, this description doesn’t match the recommendation of USA CDC, CDC recommends that the vaccine be given within 4 days from the date of exposure for the best chance to prevent onset of the disease. https://www.cdc.gov/monkeypox/considerations-for-monkeypox-vaccinations.html.

Response: Thank you for your comment. To clarify, in the previous statement (lines 315-317), we did not mention a particular time during which vaccination should be given. That being said, we added a new sentence highlighting this point as follows: “In addition, the CDC recommends that vaccination be given within 4 days from the time of exposure [42].”

Round 2

Reviewer 2 Report

The authors have addressed my concerns. Except the following minor points, I do not have any other questions.

1.       Page 1, lines 53, please update the data, just like page 5, line 115-119, “Then more than 31,000 cases were identified in 89 countries as of August 9, 2022.”

2.       Page 4, line 124-129, a Probable Case is Orthopoxvirus DNA positive by PCR, not MPXV positive. MPXV PCR positive is the confirmed case. https://www.cdc.gov/poxvirus/monkeypox/clinicians/case-definition.html#probable

3.       Page 5, line 177-178, NHPs is one of the MPXV hosts. Base on the paper that you cited, I think monkeys could not be eliminated. https://www.who.int/news-room/fact-sheets/detail/monkeypox

4.       In the “Transmission-related Preventive Measures” section, it could be better to replace “isolation” by “quarantine”.

Author Response

Cover Letter

Editor-in-Chief

Pathogens Journal

August 12th, 2022

Dear Editor,

Thank you for your letter and the opportunity to re-revise our paper on ‘Monkeypox and its Possible Sexual Transmission: Where are we now with its evidence?’ The suggestions offered by the reviewers have been immensely helpful, and we also appreciate your insightful comments on revising the structure and other aspects of the paper.

I have included the reviewer comments immediately after this letter and responded to them individually, indicating exactly how we addressed each concern or problem and describing the changes we have made. The revisions have been approved by all authors and I have again been chosen as the corresponding author. All edits that have been applied to the revised version of the manuscript have been highlighted in yellow and they can also be tracked through the “Track Changes” option on Microsoft word.

We have revised manuscript as per your comments and read to make more changes if you suggest. We thank you for your continued interest in our research.

Sincerely,

Ranjit Sah

Tribhuvan University Teaching Hospital
Institute of Medicine, Kathmandu, Nepal

[email protected]

Reviewer 2 Comments:

The authors have addressed my concerns. Except the following minor points, I do not have any other questions.

Comment 1: Page 1, lines 53, please update the data, just like page 5, line 115-119, “Then more than 31,000 cases were identified in 89 countries as of August 9, 2022.”

Response: Thank you for your comment. We updated the data in the Abstract as requested.

Comment 2: Page 4, line 124-129, a Probable Case is Orthopoxvirus DNA positive by PCR, not MPXV positive. MPXV PCR positive is the confirmed case. https://www.cdc.gov/poxvirus/monkeypox/clinicians/case-definition.html#probable

Response: Thank you for highlighting this point. We edited this part as follows: “A probable case is reported as one that meets the suspected criteria of having a positive Orthopoxvirus DNA test by PCR.”

Comment 3: Page 5, line 177-178, NHPs is one of the MPXV hosts. Base on the paper that you cited, I think monkeys could not be eliminated. https://www.who.int/news-room/fact-sheets/detail/monkeypox.

Response: Thank you for your comment. We edited this part as follows: “A wide variety of species have been reported to be susceptible to MPXV. These include tree and rope squirrels, Gambian marsupial, monkeys, non-human primates (NHPs), as well as other species [12, 17, 18]. Additionally, MPX disease is common among rodents in Africa as well [12].” We also added a citation to the WHO’s statement in this regard. Many thanks!

Comment 4: In the “Transmission-related Preventive Measures” section, it could be better to replace “isolation” by “quarantine”.

Response: Thank you for your suggestion and for your help in improving the quality of our manuscript. We removed the word “isolation” throughout this section and replaced it with “quarantine” while editing the sentence if needed to maintain its integrity.

This manuscript is a resubmission of an earlier submission. The following is a list of the peer review reports and author responses from that submission.

Round 1

Reviewer 1 Report

I really enjoyed reading this paper, first because I do share a lot of opinions.

However, I have some minor suggestions to improve the readibility of the paper.

Line 96-97: i'm not sure that the recommandations are the ones issued by the WHO in 96-96. AFAIK it was not considered as a STD with anal and vaginal sex before the 2022 outbreak.

Line 110: [in Europen countries]. You may add a word about the fact that numerous cases were returning travelers.

Line 131-132: A word about the fact that the current outbreak is closely related to the central Af. clade.

Line 155: don't you mean the historical reports?? not the current outbreak

Line 158: about the lesions, you should add a word about the current outbreak, which revealed exclusive genital lesions which is one major difference between the "historical" MKV outbreaks.  Citation https://pubmed.ncbi.nlm.nih.gov/35699601/

Paragraph starting from line 254:  please introduce the ring vaccination at the end of the "reducing transmission" section because it can definitely lower the transmission.

Line 289 : section "vaccination":

Add something like vaccination is urged to be accomplished but supplies are limited considering they are military supplies. And also we should consider that a vaccination strategy requires defining deadline with objectives in order to limit the infection rate 

Ultimately about the stigma concerns I would qualify the statement. Indeed as a physician when you speak with this population (MSM) they are really "positively" concerned about the potential spreading of the disease and they want to be actors and be vaccinated. This should be emphasized.

Good luck and very nice review.